# LayerShuffle: Enhancing Robustness in Vision Transformers by Randomizing Layer Execution Order

## Abstract

Due to their architecture and how they are trained, artificial neural networks are typically not robust toward pruning, replacing, or shuffling layers at test time. However, such properties would be desirable for different applications, such as distributed neural network architectures where the order of execution cannot be guaranteed or parts of the network can fail during inference. In this work, we address these issues through a number of training approaches for vision transformers whose most important component is randomizing the execution order of attention modules at training time. With our proposed approaches, vision transformers are capable to adapt to arbitrary layer execution orders at test time assuming one tolerates a reduction (about 20%) in accuracy at the same model size. We analyse the feature representations of our trained models as well as how each layer contributes to the models prediction based on its position during inference. Our analysis shows that layers learn to contribute differently based on their position in the network. Importantly, trained models can also be randomly merged with each other resulting in functional ("Frankenstein") models without loss of performance compared to the source models. Finally, we layer-prune our models at test time and find that their performance declines gracefully.

## 1 Introduction

While demonstrating impressive performance in many domains (Krizhevsky et al., 2012; Vaswani et al., 2017; Radford et al., 2021; Rombach et al., 2022), deep learning systems demand both extensive computational resources and tight integration of their parts. For applications at scale, they therefore increasingly require the construction of large data centers with thousands of dedicated hardware accelerators. A paradigm shift from central to decentral model inference, where loosely coupled neural networks are distributed over a number of edge devices that share the computational load of the model (Gacoin et al., 2019) therefore seems ultimately desirable. Unfortunately, current deep learning models lack the robustness necessary for such a paradigm shift.

In general, artificial neural networks (ANNs) are not robust toward pruning or replacing network layers during deployment.Similarly, changing the order of execution in-between layers without further training usually results in catastrophic losses in accuracy. Nevertheless, these properties would be desirable e.g. in distributed setups as described above, where a model is executed on a number of shared nodes in a network. This way, overloaded or malfunctioning nodes could simply be skipped in favor of other available nodes. Furthermore, malfunctioning nodes or absent nodes could simply be replaced by a similar (not the same) node, allowing for simple logistics when deploying models in practice.

Augmenting models with these properties has historically been challenging. Due to the structure of the most common types of ANNs and how they are trained through backpropagation (Linnainmaa, 1970; Werbos, 1982; Rumelhart et al., 1986), each neuron can only function by adapting to both its connected input and output neurons as well as the overall desired output of the network at training time. Furthermore, the hierarchical organization of explanatory factors is usually considered a necessary prior in deep learning, i.e. one assumes that subsequent layers extract increasingly high-level features (Bengio et al., 2013). Therefore, switching the execution orders of layers implies that layers

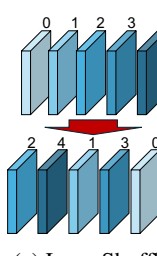 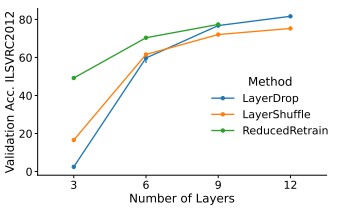 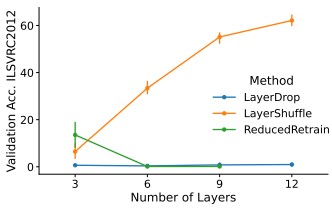

(a) LayerShuffle      (b) Pruning, original layer sequence      (c) Pruning, random layer sequence

Figure 1: *LayerShuffle training results in robust vision transformers.* (a) Illustration of the Layer-Shuffle approach. The execution order of attention modules is randomly permuted during training. (b) ImageNet2012 validation accuracy vs. number of pruned layers when executing layers in their original sequence. LayerShuffle performs similarly to LayerDrop (p=0.2), despite no layers being removed during training. (c) When additionally shuffling the layers at test time, all models fail except for LayerShuffle, whose performance degrades gracefully as more layers are removed.

would need to adapt and extract either low-level or high-level features depending on their position in the network. Unfortunately, network layers adapting in such a way to a changed order of execution appears to be infeasible for most known network architectures. The above prior is therefore violated and the overall performance of the network suffers beyond the point where the network successfully executes the task it has been trained for.

The more recently discovered transformer architecture (Vaswani et al., 2017) has been shown to be more flexible. Transformers, when trained accordingly, can be layer-pruned at test-time (Fan et al., 2019), and recent work merges similar transformer-based language models (Akiba et al., 2024), all with only moderate reduction or even an improvement in performance. We hypothesize that the reason for the high adaptability of transformers can be found in self-attention modules being able to adapt their output based on the received input. Thus it should be possible to train a transformer network to not only adapt to the variation of its input features based on the overall network input but also the variations caused by receiving input from different layers during test time.

We propose and evaluate three training approaches for vision transformers to address the robustness issues laid out above. The most important component common to all approaches is randomizing the execution order of the vision transformer's stacked self-attention-and-feed-forward modules at training time (Figure 1a). More precisely, the main contributions in this paper are:

- With LayerShuffle, the layers of a vision transformer (Dosovitskiy et al., 2020) are capable of adapting to an arbitrary execution order *at test time*, assuming one tolerates a moderate reduction in performance. Providing each layer additionally with its current position in the network improves performance only slightly compared to a model without it, suggesting that each attention layer is already capable of determining its role based on the incoming data alone.

- A UMAP analysis reveals that layers of models trained with LayerShuffle adjust their output depending on which position they hold in the network.

- Trained models can be layer-pruned at test time similar to the models trained with the techniques proposed in Fan et al. (2019), where their performance declines gracefully, i.e. models with reduced amounts of layers still remain functional.

- In addition, vision transformers, which have been made robust to execution order, can be merged with each other resulting in merged ("Frankenstein") models without loss of performance compared to the source models.

## 2 RELATED WORK

Zhu et al. (2020) find that for particular subsets of inputs, transformers perform better when changing the execution order of layers to an input-specific sequence. They optimize the execution order per sample in order to maximize the performance of the model for natural language processing tasks. While the goal in their work is to find a layer sequence of a pre-trained model that is optimal for a

given input, our approach aims to make the model robust to any sequence of execution, where layers might even be missing.

In parallel to our work on vision transformers, two groups have conducted similar experiments with the aim to understand how language models (LLMs) process data. Lad et al. (2024) found that LLMs are very robust to changing the positions of adjacent layers or ablating single layers from the model. Sun et al. (2024) perform similar experiments, and find that transformers improve iteratively upon their predictive output by subsequently refining the internal representation of the presented input. The main difference to our work is that the authors of these works do not perform any refinement on the models and switch and ablate layers locally with the aim of better understanding the inner workings of LLMs. Here we focus on methods and training approaches to increase this innate robustness of the transformer architecture to a point where models at test time function regardless of their layer execution order, and respond gracefully to the ablation of several layers in any position of the network.

Another related work is LayerDrop (Fan et al., 2019), where the authors focus on robust scalability for models on edge devices. They propose dropping whole transformer layers during training and show that this training approach allows models to still deliver acceptable (if somewhat reduced) performance upon pruning layers at test time (e.g. for balancing computational load). The main difference to our approach is that we randomly change the execution order during training, and, contrary to LayerDrop, do not remove any layers. Also, LayerDrop focuses on entirely on load balancing in compute-limited production systems while our main focus is on arbitrary execution order and the possibility to replace defective nodes by others on top of these issues in case of overloaded or malfunctioning nodes in distributed systems.

Recent work improves the performance of LLMs on predefined tasks, by merging them using evolutionary strategies (Akiba et al., 2024). Similar to Zhu et al. (2020), the authors' overall aim is to increase performance rather than robustness in distributed environments, so contrary to our approach, layer execution order and scaling for reduced numbers of layers are in general not considered.

Work on introducing permutation invariance into neural networks has been conducted by Lee et al. (2019), Tang & Ha (2021) as well as Pedersen & Risi (2022). The corresponding former two approaches exploit the permutation equivariance of attention, i.e. the fact that the order in which a sequence of vectors gets presented to the attention module does not change its result, but merely shuffles the sequence of output vectors. This equivariance is achieved by using a fixed-seed query vector in order to obtain an permutation invariant latent code. This latent code stays the same no matter in which order input tokens/patches are presented to the module. The main contrast to our work here is that we exploit permutation invariance in the order of layer executions rather than the order of tokens and patch embeddings and can therefore not make use of permutation equivariance of the attention operation, as it does not apply to switching inputs and outputs.

Finally, the work of Gacoin et al. (2019), not unlike our own, is motivated by the observation that a paradigm of distributed model inference over a number of loosely coupled compute nodes, edge devices or swarm agents promises a positive impact on the ecological and economical footprint of deep learning solutions. The authors propose a graph-theory-based framework to optimize the distribution of model parts to individual devices and optimize the overall energy consumption of the network. While our work sets out from the same motivation, it complements the approach of Gacoin et al. (2019) as the the authors do not address robustness to adverse conditions in such distributed setups while it is the entire focus of this paper. The exact distribution of our models on the other hand, is beyond the scope of our work but combining our models with the approaches in (Gacoin et al., 2019) seems a promising direction of future research.

## 3 METHODS

We investigate three approaches for arbitrary layer execution order in vision transformers (ViT; Dosovitskiy et al., 2020): First, we simply permute the order of layers randomly during training, such that every training batch is presented to the network's layers in a different random order (Section 3.1). Second, while randomly permuting the layer order as in the previous approach, we use an layer-depth encoding inspired by learned word embedding approaches (Section 3.2) to test if this additional information would further improve performance. Third, while randomly permuting layer order as in the previous approaches, we try to predict from the output of every layer at which posi-

tion the layer is currently located in the network using a small layer position prediction network for every layer (Section 3.3). A detailed overview on ViTs can be found in the appendix.

## 3.1 RANDOMLY PERMUTING LAYER ORDER DURING FORWARD PASS

During each forward pass, i.e. for each batch presented to the ViT, we randomly permute the execution order of layers during training. The intention here is to teach the layers to not only extract meaningful intermediate representations when receiving input from a particular layer, but to be able to process and encode information from and for all possible layers in the network. In terms of training, exchanging the order of layers does not require any changes in the basic error backpropagation algorithm. For the forward path, the order how weight matrices are multiplied and activation and attention functions applied changes for every batch and forward pass. This needs to be accounted in the backward pass by propagating the gradients in the precise reverse order that has been set in the forward pass, i.e. multiplying the computed per-layer gradient matrices in the correct order. As we use Pytorch (Paszke et al., 2019) in all our experiments, this aspect is taken care of the framework's autogradient feature. We refer to this model as **LayerShuffle**. To further illustrate the approach, a pseudo-code listing is given in Algorithm 1.

---

**Algorithm 1** Executing the forward path of a vision transformer with random layer order.

---

    **Input:** Input image pre-processed as a sequence $\mathbf{z}_0$,
                Sequence of $L$ vision transformer attention modules $m_1, m_2, \ldots, m_L$
    Create a new sequence $n_1, n_2, \ldots, n_L$ by randomly permuting $m_1, m_2, \ldots, m_L$,
    **for** i = 1 **to** $L$ **do**
        $\mathbf{z}_i = n_i(\mathbf{z}_{i-1})$,
    **end for**
    Return $\mathbf{z}_L$ to be post-processed by the transformer's output layer.

---

## 3.2 LAYER POSITION ENCODING

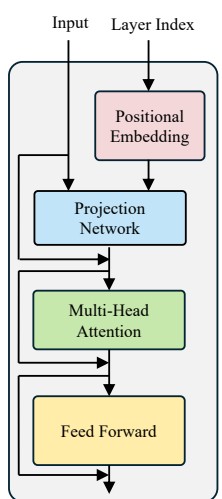

Figure 2: Attention module with layer position encoding.

In the second approach, **LayerShuffle-position**, we provide each layer with its current position in the network. Through this variation we aim to test if each layer can already adapt sufficiently by itself to information coming from different layers during test time or if giving it the current position can help further. In more detail, jointly with permuting the layer execution order, each layer learns a vector embedding $\mathbf{e}_{\text{layer}}{}^p \in \mathbb{R}^F$ for each possible index position $p \in [1, L]$ of the layer during training, where L is the number of layers and $F = 32$ is our chosen embedding dimension. The layer's current index $p$ in the network is presented together with the input to the layer $\mathbf{z}_{t-1}$ (Figure 2). The layer fetches the embedding vector $\mathbf{e}_{\text{layer}}{}^p$ associated with the passed index $p$ and concatenates it to the input vector $z_{t-1}$: $\mathbf{h}_t = \text{concat}(\mathbf{z}_{t-1}, \text{repeat}(\mathbf{e}_{\text{layer}}{}^p, N + 1))$. N is the number of patches extracted form the input image, the functions concat and repeat respectively concatenate and repeat tensors along their last (most varying) dimension. A projection network, which consists of a LayerNorm (LN) (Ba et al., 2016) module, a single linear layer $\mathbf{W}_{\text{proj}}$, a GELU (Hendrycks & Gimpel, 2016) activation function as well as a Dropout (Srivastava et al., 2014) module, is then used to combine input and embedding and reduce it again to the used latent dimension $D$ of the transformer. To ensure gradient flow during training, a residual connection is added as well:

$$\mathbf{z}_t'' = \text{Dropout}(\text{GELU}(\text{LN}(\mathbf{h}_t)W_{\text{proj}})) + \mathbf{z}_{t-1}$$

The resulting output $\mathbf{z}_t''$ is passed on to a regular multi-head-attention-and-feed-forward structure as described in Equations 1 and 2.

## 3.3 PREDICTING CURRENT LAYER POSITION

To determine if the incoming information to each attention layer is indeed sufficient for it to figure out its role, we specifically test for this ability with the **LayerShuffle-predict** variant. We equip each layer of the network with a simple position prediction module that takes the current layer output as an input and seeks to predict the current position of the layer in the network (Figure 3). The module consists of a single linear layer $\mathbf{W}_{pred} \in \mathbb{R}^{D \times L}$ receiving layer-normalized (LN)input. $\mathbf{u} = \text{LN}(\mathbf{z}_t)\mathbf{W}_{pred}$.

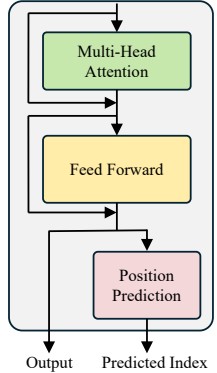

Each of these layer order prediction modules optimizes a cross-entropy loss where then the overall network optimizes the loss $\mathcal{L}_{\text{out}} + \sum^{\forall i} \mathcal{L}_i$. Here, $\mathcal{L}_{\text{out}}$ is the regular cross-entropy loss of the output layer, and $\mathcal{L}_i$ is the layer position prediction loss of layer i, which is also a cross-entropy loss:

$$\mathcal{L}_i = -\log\left(\frac{\exp(u_p)}{\sum^{\forall l \in L} \exp(u_l)}\right),$$

Figure 3: Attention module with layer position prediction.

where $L$ is the number of layers in the network, $\mathbf{u}$ is the $L$-dimensional output of the position prediction network of layer $i$, and $u_l$ denotes the $l$-th dimension of the vector. $u_p$ is the output logit denoting the network's predicted confidence that the layer currently is deployed at its actual position with index $p$.

## 4 EXPERIMENTS

We conduct our experiments on the ILSVRC2012 dataset (Russakovsky et al., 2015), more commonly termed ImageNet2012, as well as the CIFAR-100 dataset Krizhevsky et al.. We use the original *ViT-B/16* (Dosovitskiy et al., 2020) vision transformer, as well the *DeiT-B* distilled data-efficient image transformer Touvron et al. (2021). Pre-trained weights for ImageNet2012 are publicly available for both models (Dosovitskiy et al., 2020; Wu et al., 2020; Touvron et al., 2021).

The *ViT-B/16* has been pre-trained on ImageNet21k (Deng et al., 2009; Ridnik et al., 2021) at an 224×224 input image resolution and refined on ImageNet2012 at the same resolution. *DeiT-B* has the same architecture as *ViT-B/16*, but uses an additional destillation token during training, which is used to distill the inductive bias of a large convolutional network into the transformer in order to require less training data. It is pre-trained exclusively on ImageNet2012.

Both models are again refined on both ImageNet2012 and CIFAR-100 at the same resolution, but using the training processes as described in Section 3. That is, layer execution order is randomly permuted while refining the model. To establish a baseline, on ImageNet2012, we refine the original models for one more epoch on without changing the layer order. Any longer training was found unlikely to bring additional improvement in preliminary experiments since our networks are already pretrained on ImageNet. For CIFAR-100, we refine our baseline for 20 epochs. For each approach, including the baselines, we train 5 networks and compare their average validation accuracy.

All models are refined using Adam (Kingma & Ba, 2014) ($\beta_1 = 0.9$, $\beta_2 = 0.999$, $\epsilon = 10^{-6}$), where an initial learning rate of $10^{-4}$ was empirically found to work best. In terms of batch size, we evaluate training batch sizes of 640 images, which is the maximum multiple of 8 that can fit in the video memory of our used GPU, as well as 128 images for models that benefit from a smaller batch size. Even smaller batch sizes do not yield any improvement in performance for our models. Inspecting training curves shows that for ImageNet2012 the performance of models plateaus at 20 epochs the latest, which is therefore set as the maximum number of training epochs. For CIFAR-100 we use 100 epochs since the models were pretrained on a different dataset, i.e. ImageNet. We use a form of early stopping by evaluating the model achieving the lowest crossentropy loss on the validation set after the maximum amount of training epochs. All models have been trained on a single NVIDIA H100 Tensor Core GPU with 80GB of memory. Training a single model on ImageNet2012 for 20 epochs takes about 7 hours whereas CIFAR-100 training times are significantly shorter due to the smaller train set.

## 4.1 Sequential vs. arbitrary execution order

The average accuracy for all approach on all vision transformer architectures and datasets is shown in Table 1. We make the following observations across all models and datasets:

On both the CIFAR-100 and the ImageNet2012 dataset, our baselines refined from pre-trained *ViT-B/16* and *DeiT-B* models perform very much as expected. For a classic sequential execution order of the model layers, on ImageNet2012 the trained models achieve an average validation accuracy very close to the performance of the respective original pretrained models (Dosovitskiy et al., 2020; Touvron et al., 2021). Our refined baseline *ViT-B/16* obtains an average accuracy of 82.61% with a standard deviation of 0.08. The *DeiT-B* model attains a slightly lower accuracy of 81.16% with a standard deviation of 0.06. Baseline results CIFAR-100 look similar with *ViT-B/16* and *DeiT-B* achieving 89.62% (standard deviation: 0.26) and 87.31% (standard deviation: 0.25) respectively. Not surprisingly, for an arbitrary layer execution order, the average model accuracy declines catastrophically to below 1% for all trained models on both datasets. Our original assertion that in general, ANNs are not robust to changing the execution order of their layers, is in line with these results.

Our LayerShuffle approaches show slightly lower performance than the baselines when executing layers in their original order. On ImageNet2012, our trained *ViT-B/16* models obtain average accuracies of 75.22, 75.28, and 74.41 respectively for our *LayerShuffle*, *LayerShuffle-position* and *LayerShuffle-predict* approaches. Average accuracies for *DeiT-B* models are in a similar range with 76.57, 76.18, and 75.08 for our respective *LayerShuffle*, *LayerShuffle-position* and *LayerShuffle-predict* approaches. On CIFAR-100 on the other hand the gap between baseline and LayerShuffle models is somewhat larger for the trained *ViT-B/16* models. These models merely achieve 64.43, 61.98 and 60.88 with slightly higher standard deviations for *LayerShuffle*, *LayerShuffle-position* and *LayerShuffle-predict* approaches. For *DeiT-B* models on the other hand, these approaches perform similarly well to the models trained on ImageNet with scores of 75.04, 72.13 and 66.94 for the above mentioned techniques. A possible explanation for these discrepancies could be found in *ViT-B/16* models requiring more and more diverse training data compared to *DeiT-B* models, which has been pre-trained with the aim to reduce the amount of required training data.

Despite being outperformed by the baseline in a sequential execution order setting, all models improve dramatically over their corresponding baseline models in an arbitrary execution order setting. Taking a closer look at LayerShuffle model performance in that setting, we find that the simplest approach performs very well across both architectures and datasets. For both ImageNet2012 as well as CIFAR-100 validation sets *DeiT-B* trained on LayerShuffle yields the best performance with average accuracies of 66.62 and 64.99 respectively, narrowly outperforming *LayerShuffle-position*, which receives information about the layer position. *LayerShuffle-position* achieves scores of 66.62 and 64.99 on these datasets. For *ViT-B/16* models on the other hand, *LayerShuffle-position* outperforms *LayerShuffle*. The former achieves scores of 63.61 and 55.47 of ImageNet2012 and CIFAR-100, with the latter performing only slightly worse with 62.77 and 54.34. The most likely explanation for models of the *ViT-B/16* architecture achieving significantly lower accuracies on CIFAR-100 than *DeiT-B* models can again be found in the former requiring less training data than the latter.

We find that the position prediction approach, *LayerShuffle-predict* is outperformed by both our remaining approaches on all datasets and architectures. On ImageNet2012, refined *ViT-B/16* models achieve average accuracies of 61.18 whereas *DeiT-B* models attain 64.51. On CIFAR-100 the former score 53.53, the latter 58.77. A possible explanation might be that due to optimization of multiple objectives (fitting both the output labels as well as predicting the current position of the layer) this approach requires more careful hyperparameter tuning.

A further interesting observation is to be made when comparing the performance for sequential and arbitrary execution order for each approach respectively. For all approaches, using the original layer order for sequential execution still performs better than an arbitrary order. This is most likely a consequence of fine-tuning from a sequentially trained model.

For the layer position prediction approach, we measure the average accuracy of layer position predictions over all five trained *LayerShuffle-predict* models, and find that the layer position is predicted correctly in 99.99% of all cases. These results demonstrate that each layer has enough information coming from its inputs alone to predict where it is in the network, providing the basis to adapt to its current position. We investigate this further when analyzing intermediate network representations in Section 4.3. In conclusion, refining a pre-trained model while randomly permuting the execution

order of the network layers can make a model more robust towards such arbitrary execution orders at test time. On the other hand, Dropout and LayerNorm by themselves do not have the same effect and fail to produce networks robust against layer shuffling.

Table 1: *Approach accuracy for sequential and arbitrary execution order of layers on the ImageNet2012 (IN2012) and CIFAR-100 (CIFAR) validation sets.* The baseline models perform best when executed sequentially but fail catastrophically when the layers are executed in an arbitrary order. Even the simplest LayerShuffle variant, in which the model does not have any information about its current position, reaches accuracies above 60 percent. All of our proposed training approaches permit the models to be executed with arbitrary layer execution order at test time, while still delivering good performance for the original model execution order.

|  | model | layer order | baseline | LS | LS-pos | LS-pred |
|---|---|---|---|---|---|---|
| IN2012 | ViT-B/16 | sequential | $\mathbf{82.61}\pm0.08$ | $75.22\pm0.28$ | $75.28\pm0.18$ | $74.41\pm0.20$ |
| IN2012 | ViT-B/16 | arbitrary | $0.13\pm0.03$ | $62.77\pm0.41$ | $\mathbf{63.61}\pm0.23$ | $61.18\pm1.06$ |
| IN2012 | DeiT-B dist. | sequential | $\mathbf{81.16}\pm0.06$ | $76.57\pm0.12$ | $76.18\pm0.46$ | $75.08\pm0.27$ |
| IN2012 | DeiT-B dist. | arbitrary | $0.12\pm0.05$ | $\mathbf{66.62}\pm0.4$ | $65.89\pm1.28$ | $64.51\pm0.85$ |
| CIFAR | ViT-B/16 | sequential | $\mathbf{89.62}\pm0.26$ | $64.43\pm0.95$ | $61.98\pm0.49$ | $60.88\pm0.76$ |
| CIFAR | ViT-B/16 | arbitrary | $0.64\pm0.13$ | $54.34\pm2.64$ | $\mathbf{55.47}\pm1.41$ | $53.53\pm1.45$ |
| CIFAR | DeiT-B dist. | sequential | $\mathbf{87.31}\pm0.25$ | $75.04\pm0.83$ | $72.13\pm1.5$ | $66.94\pm0.6$ |
| CIFAR | DeiT-B dist. | arbitrary | $0.53\pm0.16$ | $\mathbf{64.99}\pm0.76$ | $64.31\pm1.70$ | $58.77\pm1.18$ |

## 4.2 REMOVING LAYERS DURING TEST TIME

To determine how neural networks trained with LayerShuffle would perform when several devices in a (distributed) model become unavailable, we further investigate the effect of pruning an increasing amount of layers during test time. We evaluate its average validation accuracy over 5 models when only using 3, 6, or 9 layers. In addition, we refine the original *ViT-B/16* transformer using LayerDrop (Fan et al., 2019) with a drop probability of 0.2 (as recommended by the authors) and compare it as a baseline to our approach under identical conditions. Note that whenever we evaluate the accuracy of our proposed approaches as well as the baseline, we do so two times: Once, for the original "sequential" layer order as originally intended and trained for the *ViT-B/16* transformer, and once with arbitrary layer execution order where we change the order randomly for every forward path.

For sequential execution (Figures 1b), LayerDrop with a drop rate of 0.2 behaves similarly to LayerShuffle, with the exception that our approach performs better for a small number (3) of layers with an average accuracy of approximately $18\%$ vs. close to $0\%$ for LayerDrop. While for 6 layers, both approaches are roughly on par, for 9 layers LayerShuffle is slightly outperformed by LayerDrop as both approaches show an average accuracy in the $70-80\%$ range. At the full amount of 12 layers, this gap in average accuracy stays roughly the same as the LayerDrop-refined model closes in on the full accuracy of the original model, while our LayerShuffle approach achieves slightly lower accuracies (see also Table 1). For comparison, we also visualize models where we refined a reduced number of 3, 6, and 9 layers: while delivering similar performance as LayerDrop for 9 and 12 layers, these models perform significantly better than the previously discussed approaches at lower numbers, i.e. 3 and 6 layers. They do however, bear the drawback that for each specific amount of layers a new model must be refined from the original model, whereas for both LayerDrop and our LayerShuffle approach, only a single full-size model needs to be refined and the number of layers can be configured at will at test time.

For arbitrary execution (Figure 1c), LayerShuffle is the only approach that succeeds, with the average accuracy improving as the number of layers is increased. LayerDrop does not perform well regardless of the number of layers in the model. A noteworthy detail is the comparable high average accuracy of the fully retrained baseline with 3 layers. Given the low performance of the refined models with 6 and 9 layers, as well as that there are only 6 possible permutations for 3 layers, the most likely explanation is that one of the 5 random permutations evaluated for the model was the original layer execution order the model has been trained for, i.e. [1,2,3] therefore skewing the achieved accuracy in this case. In conclusion, we find that our proposed approach has similar test-time scaling capabilities as LayerDrop, while still ensuring robustness towards arbitrary layer execution orders.

## 4.3 ANALYSIS OF INTERMEDIATE NETWORK REPRESENTATIONS

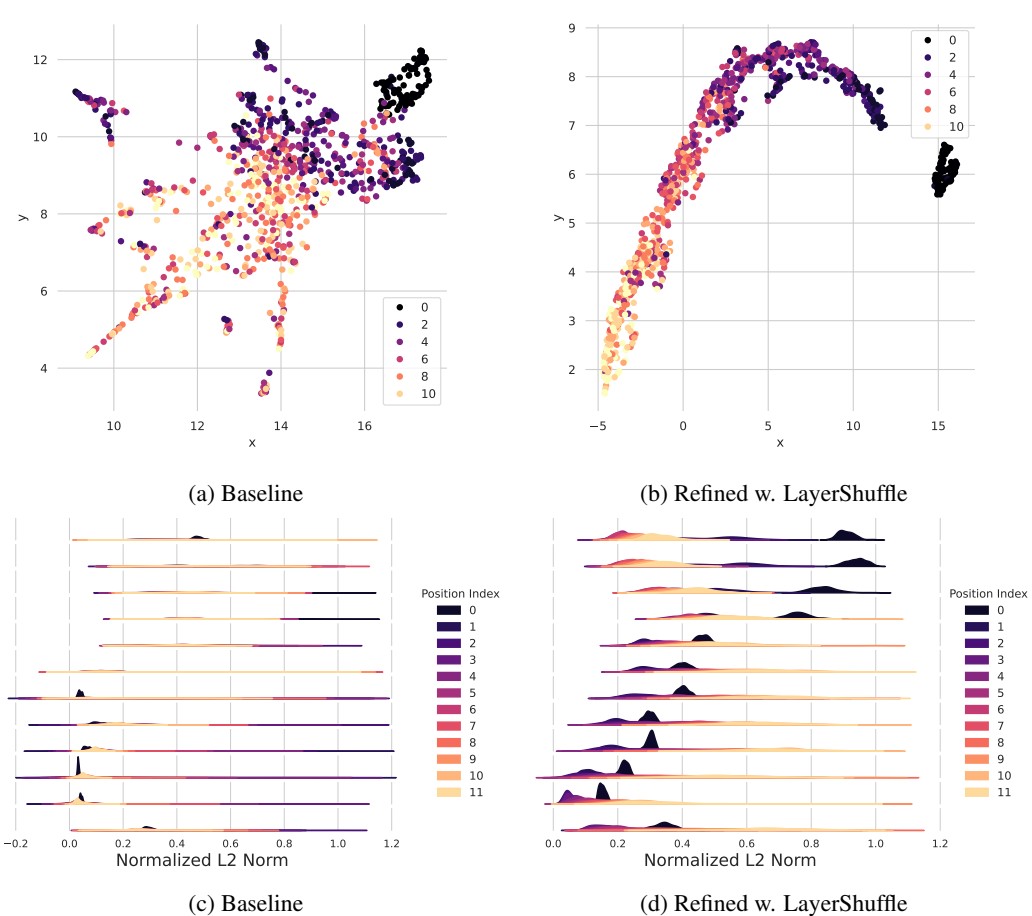

(a) Baseline

(b) Refined w. LayerShuffle

(c) Baseline

(d) Refined w. LayerShuffle

Figure 4: *UMAP-projected embeddings and contributions to model prediction (estimated distribution of normalized L2 norms of class token) of layer outputs trained with shuffling execution order, baseline for comparison.* Contrary to the baseline (a), the layer for a *LayerShuffle*-trained network (b) produces outputs in different subspaces of the latent space depending on their current position in the network. Darker colors indicate layer positions closer to the input; layer positions close to the output are shown in light colors. While layers in the baseline model overall contribute equally to the predictive output of the model, regardless of their current position in the network (c), the contribution of layers in the *LayerShuffle*-trained model's prediction (d) varies based on the distance to it's original position in the networks. Refinement of the model conditions its layers to only contribute to the overall predictive output if the received input lies within the layers learned distributions of inputs.

To gain a deeper insight into how information is encoded in the models, we conduct two experiments. First, we compute Uniform Manifold Approximation and Projection (UMAP) (McInnes et al., 2018) embeddings of the entire output of a particular attention module (i.e. combined self-attention and feed-forward layers), where we color-code all output vectors based on the position the module held in the network when producing this output. In more detail, we concatenate all patch tokens of a single image together with the class token as a single vector, and use this representation as a single state vector in our compression. To extract a sufficient number of these state vectors, we present 1,000 randomly sampled images from the ImageNet2012 validation set to a LayerShuffle-trained model. While we use an evaluation batch size of 1 image and record all outputs of a single, previously selected layer, we randomly permute the execution order of layers such that the selected

layer changes position in the network during every forward path. After the layer's output vectors for all 1,000 images have been recorded, a UMAP reduction of the output space to 2D is performed.

Second, in order to investigate how much each layer contributes to the final classifier output when deployed in different positions within the model, we compute the L2-Norm of the class-token of each layer output. We correct for the contribution of previous layers by subtracting the class token of the previous layer before computing the token's norm. That way, we consider solely the additive contribution of the layer to the class prediction of the model. We collect these token norms for all network layers as we shuffle their position while presenting 1000 randomly sampled input images in an identical manner as in the previous experiment.

Finally, to establish a baseline, we extract both representations for the original *ViT-B/16* weights Dosovitskiy et al. (2020) as well. Figure 4b shows the obtained visualizations for the original *ViT-B/16* model acting as a baseline as well as our model refined with *LayerShuffle*. In more detail, Figures 4a and 4b show the UMAP embeddings of a single layer's output for both the baseline and our model. The current position of the layer in the network when producing a given output is color-coded from dark (position close to the input) to light (position close to the output). Note that this information about the layer position has not been presented to the UMAP algorithm. Apart from rough trends, no clear ordering of the space is visible for the baseline (Figure 4a). For LayerShuffle, while there is no sharp separation between outputs generated at different positions in the network, the layer clearly adapts to its current position and extracts different features for different positions in the network (Figure 4b).

A further interesting observation is the very distinct collection of points for layer positions close to the input, which are detached from the remaining manifold of points. This results suggests that extracting low-level features, requires special treatment. Figures 4c and 4d show the distributions on the normalized L2-norm of additive contributions to class tokens for different layer positions in the transformer for both the baseline and our model. Each x-axis in the plot corresponds to a single layer of the network, where position of the layer in the network is color-coded again from close to the input (dark) to close to the output (light). x-axes are also ordered corresponding to the layer's original position in the pre-trained model, where the order of layers is top to bottom. We can see that for the baseline model norms are basically spread out over the whole range. This implies that layers in the baseline model overall contribute evenly to the predictive output of the model, regardless of their current position in the network.

On the other hand, the ridge plot gathered from layer outputs of the model refined with our method paints a different picture. The norm of attention modules output and therefore it's contribution to the model's prediction varies based on the distance to it's original position in the networks. Modules which were originally closer to the input (x-axes on top of the plot) often show larger contributions to the predictive output of the model when on positioned closer to the input and vice versa. This indicates that our refinement of the model conditions its layers to contribute to the overall predictive output if the received input lies within the layers learned distributions of inputs (i.e. the layer is close at a position assigned to it in the original pre-trained network), and withhold or reduce their output otherwise. This is also in line with recent work conducted in parallel Sun et al. (2024), which frames transformer layers as incrementally refining a rough sketch of the model's output, an iterative process which is enabled by the transformer's extensive utilization of skip-connections.

In conclusion, our analysis indicates that refining networks with *LayerShuffle* makes vision transformers robust to arbitrary execution orders as it trains the layers to solely add to the models contribution if the layer input is in-distribution and reduce their output otherwise, in which case the model's skip-connection forwards the out-of-distribution output to the subsequent layer.

## 4.4 MERGING MODELS WITH ARBITRARY EXECUTION ORDER

Being robust against permuting the layer execution order, opens interesting other possibilities such as model merging, i.e. creating a new model from the layers of several identically trained models. The underlying rationale is that such merged models could also occur in a distributed setting, where compute nodes, whose layers have been trained as part of distinct models, but with the same training process, could form ad-hoc models together.

To construct merged models, where each layer stems from a different model, we require 12 models for the 12 layers of the *ViT-B/16*. We therefore train 7 more networks for our *LayerShuffle* approach and the baseline. Subsequently, we create 100 merged models (out of 12! possible combinations) by

randomly sampling from these models for our proposed approach as well as the baseline respectively (models are not mixed between approaches). As mentioned previously, layers are sampled in such a way that no two layers in a merged model stem from the same model. We then evaluate the validation accuracy of all 100 models for both approaches.

Table 2 summarises the results. The merged baseline model *ViT-B/16* deteriorates from $82.61\%$ average accuracy to $1.87\%$ (despite sequential layer execution as required by the model) making the resulting merged model effectively unusable. The merged *LayerShuffle* models, on the other hand, perform slightly below the original model with an average accuracy of $59.68\%$ as opposed to the $62.77\%$ of the latter. Less surprisingly, merged models show a higher standard deviation at $1.15$ percentage points for the merged models vs. $0.41$ percentage points for the original ones as merged models do not contain any two layers that have been trained together, which makes their performance vary more. We can further improve performance by ensembling the 12 models trained with LayerShuffle, using the average of their output logit vectors. Such neural network ensembles often reach a better performance (Hansen & Salamon, 1990), which is also the case here with a significant improvement and an accuracy of $69.26\%$ for LayerShuffle Ensemble.

In conclusion, we find that permuting layer order during training enables the construction of merged (or "Frankenstein") vision transformers, where each layer of the transformer can be taken from a different model, as long as all models have been refined from the same base model on the same data.

Table 2: *Validation accuracy of merged ViTs on ImageNet2012.* Merged *LayerShuffle* achieve an accuracy close to the average accuracy of the original models, while for the baseline, the merged model exhibits very low accuracy. Ensembles of *LayerShuffle* models show clear improvement over single models.

| Model | Top-1 Acc. | Top-5 Acc. | layer order |
|---|:---:|:---:|:---:|
| ViT-B/16 merged | $1.87 \pm 6.51\%$ | $4.53 \pm 11.82\%$ | sequential |
| LayerShuffle merged | $59.68 \pm 1.15\%$ | $82.16 \pm 1.03\%$ | arbitrary |
| LayerShuffle Ensemble | $69.26\%$ | $88.76\%$ | arbitrary |

## 5  DISCUSSION AND FUTURE WORK

This paper presented a new approach called LayerShuffle, which enabled vision transformers to be robust to arbitrary order execution, pruning at test time, as well as adhoc-construction of merged models. For sequential execution, LayerShuffle performs on average only slightly worse than the LayerDrop approach but is the only method that works when the layer execution is arbitrary. Our analysis confirmed that layers of models trained with LayerShuffle adjust their output depending on which position they hold in the network. Furthermore, our results indictate that refining networks with *LayerShuffle* trains the layers to only contribute to the model's class prediction if the layer input is in-distribution and reduce their output otherwise, in which case the layer's skip-connection forwards the barely modified out-of-distribution embedding to the subsequent layer.

Finally, we investigated whether it is possible to build merged models from the models trained with LayerShuffle and found the performance of the built merged models to be only slightly less than the performance of our trained models, contrary to the baseline, where virtually all merged models delivered very poor performance.

In the future, these properties could make *LayerShuffle*-trained models ideal candidates to be distributed over a number of very loosely coupled compute nodes to share the computational load of model inference. Given the enormous engineering, financial and logistical effort as well as the environmental impact (Strubell et al., 2020) of building and maintaining datacenters for state-of-the-art deep learning approaches on the one hand, as well as the large amount of available, but scattered compute through existing smartphones, laptop computers, smart appliances and other edge devices on the other hand, approaches that allow ad-hoc neural networks performing inference together could be of great impact. We therefore consider the deployment and orchestration of our trained models onto an actual set of edge devices and the practical implementation of the inference process on a network of such devices, likely by combining our approach with previously proposed frameworks to address this issue (Gacoin et al., 2019), a very promising direction of future research.

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

$$\text{MSA}(\mathbf{z}) = [\text{SA}_1(\mathbf{z}); \, \text{SA}_2(\mathbf{z}); \, \dots; \, \text{SA}_k(\mathbf{z})]\mathbf{U}_{\text{msa}}$$

In an attention module the multi-head self-attention layer is followed by a multi-layer-perceptron (MLP) layer transforming the recently combined embeddings to extract new feature representations. Before presenting $\mathbf{z}$ to each layer in the module, the embeddings are normalized using LayerNorm (Ba et al., 2016). To ensure consistent gradient flow during training, residual connections (He et al., 2016) are behind both the MSA and the MLP layers (Wang et al., 2019). Furthermore, as a regularization measure, Dropout (Srivastava et al., 2014) is applied after every MSA and MLP layer. In summary, given the sequence $\mathbf{z}_{t-1}$ from a previous attention module as input, we first compute the intermediate representation

$$\mathbf{z}_t' = \text{MSA}(\text{LN}(\mathbf{z}_{t-1})) + \mathbf{z}_{t-1}, \tag{1}$$

which is the presented to the MLP layer to compute the final output of the module

$$\mathbf{z}_t = \text{MLP}(\text{LN}(\mathbf{z}_t')) + \mathbf{z}_t'. \tag{2}$$

Finally, after N attention modules, the first vector of the sequence (corresponding to the `class`-token in the preprocessed input) is handed to a linear layer $\mathbf{W}_{\text{out}} \in \mathbb{R}^{D \times C}$ to predict the final class of the image: $\mathbf{y} = \text{argmax}(\mathbf{z}_L^0 \mathbf{W}_{\text{out}})$. $C$ denotes the number of classes.

