# OpenReview forum: "LayerShuffle: Enhancing Robustness in Vision Transformers  by Randomizing Layer Execution Order"
_ICLR.cc/2025/Conference — ICLR 2025 Conference Withdrawn Submission_

### Official Review · Reviewer_FWeg · 2024-10-27

**Soundness:** 3
**Presentation:** 4
**Contribution:** 3
**Rating:** 6
**Confidence:** 4

**Summary:**

This paper tackles the challenge of layer-order robustness in vision transformers by proposing a training method that randomizes attention module execution order.
This approach allows the model to handle arbitrary layer orders at test time with only a slight reduction in accuracy.
Through analysis, the authors find that layers adapt based on their inference position, contributing variably to the model’s predictions
Additionally, the trained models can be combined to create effective "Frankenstein" networks without compromising performance.
Finally, they demonstrate that the models maintain graceful performance degradation under layer pruning, suggesting strong potential for distributed architectures where execution order may be unreliable.

**Strengths:**

S1. The methodology is highly intuitive and straightforward, making it easy to comprehend. The clarity in the writing style further enhances readability.

S2. The proposed method demonstrates strong potential for real-world applications and provides a promising foundation for future research directions.

S3. The paper presents a solid level of technical novelty, with innovative thinking underpinning the approach.

**Weaknesses:**

W1. The study primarily evaluates a low-level task, specifically classification, which raises the question of whether similar performance trends would hold for more complex tasks. It would be helpful if the authors could provide results for additional tasks, such as segmentation or regression, to gauge the method’s broader applicability.

W2. While the evaluation results on ImageNet show an acceptable performance drop relative to the baseline, Table 1 indicates a more substantial performance drop on the CIFAR-100 dataset. This suggests that performance may correlate with task complexity, though the current study does not provide sufficient insight into this relationship.

W3. I find it challenging to conceive of grounded examples for S2. This may be due to my limited expertise in distributed systems. Could the authors provide examples illustrating practical applications of the approach? Specifically, how does the robustness in arbitrary layers contribute to these applications?

W4. Some captions lack sufficient explanatory information. For instance, in Figure 1(b), it’s unclear what “ReducedRetrain” represents—is it the original model? Additionally, in Table 1, does the italicized font indicate the second-best performance?

**Questions:**

Since I have included all questions and concerns in the weaknesses section, here are some minor corrections and typos I noticed:

1. Line 41-42: Change "deployment.Similarly, ..." to "deployment. Similarly, ..."
2. I think it would be beneficial to include a quantitative comparison with the baseline LayerNorm in Table 1, rather than only presenting it as a graph in Figure 1.

---

### Official Review · Reviewer_tPkq · 2024-11-02

**Soundness:** 2
**Presentation:** 2
**Contribution:** 2
**Rating:** 1
**Confidence:** 5

**Summary:**

This paper proposes a new layer shuffling method for Transformers to spread the model's components onto different computing devices so that the order of layers does not matter and immense speedup can be achieved. Also, the proposed method claims to be able to combine randomly trained networks without loss of accuracy.

**Strengths:**

1. The paper method is well written.
2. The concept is interesting.

**Weaknesses:**

1. The concept is interesting; however, the argument that the computing machine can fail at a time does not convince me to develop this method. The paper has weak motivation.
2. The paper evaluation is not sufficient.
3. There are several Transformer architectures which have not been evaluated. Currently, the evaluation is only done on ViT/DeiT, which is not convincing to me.
4. Hyperparameter study is not done.
5. Accuracy loss is significant (20%). This accuracy gap is very substantial.
6. I have particular concerns about Figure 1c. It is unfair to compare layer-drop or reduced depth with layer shuffle configuration at deployment because the two baselines are not trained with layer shuffling order. This figure is entirely misleading and should be dropped.
7. The only change made in the transformer block is the layer index, which is not a significant contribution.
8. Please include several recent transformer-based baselines such as Swin, EfficientVit, etc.

**Questions:**

See weakness.

---

### Official Review · Reviewer_pZqV · 2024-11-02

**Soundness:** 3
**Presentation:** 3
**Contribution:** 3
**Rating:** 5
**Confidence:** 5

**Summary:**

The paper proposes a method termed LayerShuffle to shuffle the layer order of a model during training so that the model is robust to different layer orders. A UMAP analysis reveals that layers of models trained with LayerShuffle adjust their output depending on which position they hold in the network.

**Strengths:**

There are some strengths in the paper:
* The authors propose a method termed LayerShuffle to shuffle the layer order of a model during training. With LayerShuffle the model is less sensitive to the layer order and therefore more robust.
* A UMAP analysis reveals that layers of models trained with LayerShuffle adjust their output depending on which position they hold in the network.
* With LayerShuffle the model is still functional with performance degradation.

**Weaknesses:**

Although there are some spotlights in the paper, there are still some unclearnesses that need to be clarified:
* Novelty is limited. The key idea of the paper is to shuffle the layer order of a model during training so that the model is robust to different layer orders.
* Model performance drops. Line 285 says the model performance with LayerShuffle is “slightly” lower than the one without LayerShuffle. However, the degradation is over 28% on the simple classification task according to Table 1 (CIFAR, sequential, LS-pred).
* LS and LS-Pos are almost the same according to Table 1. LS-pred is always the worst. Is it necessary to have three variants without performance improvement?
* More experiments on different model architectures. It would be great if the authors could conduct experiments with 2 more sota models.
* More experiments on different tasks. It would be great if the authors could conduct experiments on 2 more tasks.
* More experiments on different datasets. It would be great if the authors could conduct experiments on 2 more datasets.

**Questions:**

Please refer to the questions in the weaknesses section.

**Details Of Ethics Concerns:**

No concerns.

---

### Official Review · Reviewer_hAiD · 2024-11-04

**Soundness:** 2
**Presentation:** 2
**Contribution:** 1
**Rating:** 5
**Confidence:** 4

**Summary:**

This paper investigates alternative training approaches for vision transformers where the layers are permuted randomly to make them robust to pruning and shuffling at test time. Variants also include using layer-depth encoding and layer position prediction.

**Strengths:**

It makes the vision transformers capable of adapting to arbitrary layer execution orders at test time.

**Weaknesses:**

I carefully reviewed the paper, and rather than addressing an extensive list of less critical issues, I will concentrate on the most significant concerns that influence my rating. My focus will be on the following two points:

1) The reduction in performance, which is around or above 20%, is substantial. This limits the useability and effectiveness of the method, making motivation unclear. In what practical real-world application is it preferable to run vision transformers, in particular the ones used in the paper (ViT and DeiT) in distributed setups where one could tolerate 20% loss? Energy consumption optimization does not seem to be a motivator since there is an added cost of data transfer across nodes in a distributed setup. Instead of layer shuffling, layer dropping or just using smaller models might be more advantageous alternatives. Why would one use "any sequence of layer execution"?

2) Robustification to changing the order of execution and shuffling has been investigated in the past, for instance, as discussed in the paper by instance-wise layer reordering (ICLR 2020), permutation invariance (many works here), LayerDrop (ICLR 2020), RandConv (ICLR 2021), and ShuffleTransformer (last two can be added to related work). As mentioned, auto-regressive models leveraging self-attention across tokens, including LLMs, are very robust to changing the positions of adjacent layers or the underlying layers from the model. After patch embedding, ViT also follows a similar architecture. Besides,
a recent work, ShuffleMamba discusses a similar stochastic layerwise shuffle regularization in training. Considering this, the novelty of the method becomes borderline.
I am open to reconsidering my rating based on a satisfactory rebuttal from the authors.

**Questions:**

See the weaknesses.

---

### Note · Authors · 2024-11-25

**Comment:**

Dear reviewers, thank you for taking the time to review our manuscript and providing insightful feedback. Based on your feedback as well as further experiments from our side, we feel there is still room for our work to evolve and mature. We have therefore decided to retract our manuscript and resubmit it on a later point.
Kind regards,
the authors

**Withdrawal Confirmation:**

I have read and agree with the venue's withdrawal policy on behalf of myself and my co-authors.